# Gluten Degrading Enzymes for Treatment of Celiac Disease

**DOI:** 10.3390/nu12072095

**Published:** 2020-07-15

**Authors:** Guoxian Wei, Eva J. Helmerhorst, Ghassan Darwish, Gabriel Blumenkranz, Detlef Schuppan

**Affiliations:** 1Department of Molecular and Cell Biology, Henry M. Goldman School of Dental Medicine, 700 Albany Street, Boston, MA 02118, USA; weigx@bu.edu (G.W.); histatins@gmail.com (E.J.H.); gdarwish@bu.edu (G.D.); gaboblumenkranz@hotmail.com (G.B.); 2Institute for Translational Immunology and Research Center for Immunotherapy (FZI), Johannes Gutenberg University (JGU) Medical Center, 55131 Mainz, Germany; 3Division of Gastroenterology, Beth Israel Deaconess Medical Center, Harvard Medical School, Boston, MA 02215, USA

**Keywords:** celiac disease, gluten, enzyme therapy, treatment, autoimmunity, endopeptidase, glutenase, wheat, enteric coating

## Abstract

Celiac disease (CeD) affects about 1% of most world populations. It presents a wide spectrum of clinical manifestations, ranging from minor symptoms to mild or severe malabsorption, and it may be associated with a wide variety of autoimmune diseases. CeD is triggered and maintained by the ingestion of gluten proteins from wheat and related grains. Gluten peptides that resist gastrointestinal digestion are antigenically presented to gluten specific T cells in the intestinal mucosa via HLA-DQ2 or HLA-DQ8, the necessary genetic predisposition for CeD. To date, there is no effective or approved treatment for CeD other than a strict adherence to a gluten-free diet, which is difficult to maintain in professional or social environments. Moreover, many patients with CeD have active disease despite diet adherence due to a high sensitivity to traces of gluten. Therefore, safe pharmacological treatments that complement the gluten-free diet are urgently needed. Oral enzyme therapy, employing gluten-degrading enzymes, is a promising therapeutic approach. A prerequisite is that such enzymes are active under gastro-duodenal conditions, quickly neutralize the T cell activating gluten peptides and are safe for human consumption. Several enzymes including prolyl endopeptidases, cysteine proteases and subtilisins can cleave the human digestion-resistant gluten peptides in vitro and in vivo. Examples are several prolyl endopeptidases from bacterial sources, subtilisins from *Rothia bacteria* that are natural oral colonizers and synthetic enzymes with optimized gluten-degrading activities. Without exception, these enzymes must cleave the otherwise unusual glutamine and proline-rich domains characteristic of antigenic gluten peptides. Moreover, they should be stable and active in both the acidic environment of the stomach and under near neutral pH in the duodenum. This review focuses on those enzymes that have been characterized and evaluated for the treatment of CeD, discussing their origin and activities, their clinical evaluation and challenges for therapeutic application. Novel developments include strategies like enteric coating and genetic modification to increase enzyme stability in the digestive tract.

## 1. Introduction

Celiac Disease (CeD) is the most common food-induced heritable and life-long inflammatory disease in humans. CeD is triggered upon ingestion of wheat gluten or similar proteins found in other cereals such as barley (hordeins) and rye (secalins). Histologically, CeD is characterized by flattening of the proximal intestinal villi and crypt hyperplasia resulting in a loss of resorptive surface area. This frequently causes malabsorption of nutrients, vitamins and minerals and increases the risk of anemia, osteoporosis, infertility, otherwise rare small intestinal cancers and a wide spectrum of autoimmune diseases. CeD and the associated autoimmune diseases are linked to common and necessary genetic predisposition, human lymphocyte antigen (HLA-DQ2 or -DQ8 that are required for antigenic presentation of certain gluten peptide to gluten reactive T cells in the intestine. This drives an inflammatory T helper 1 (Th1) cell response resulting in villous atrophy and usually in clinical disease. Approximately 1% of most of the world’s populations is affected by CeD, while HLA-DQ2 and HLA-DQ8 occur in most affected populations, suggesting that a spectrum of other, environmental and genetic factors contribute to CeD manifestation. Notably, the majority of patients with CeD remains undiagnosed, despite the availability of a highly specific blood test for active CeD [1,2,3,4]. One reason for this is that nowadays most patients with active CeD have no or only minor abdominal symptoms such as diarrhea or overt malabsorption, despite concomitant extraintestinal autoimmunity and an increased risk for malignancy. Once diagnosed, a strict gluten-free diet (GFD) is the only effective measure to treat CeD and helps to ameliorate or prevent the development of its comorbidities. However, the GFD is often difficult to maintain in our societies or during travels, and many diet compliant patients do not achieve remission or continue to have symptoms [5]. Therefore, non-dietary (adjunctive) therapies for CeD are urgently needed [3,6,7]. In this review, we discuss the origin, structure and protease sensitivity of gluten and its immunogenic peptides that are the critical molecules that elicit the inflammatory intestinal T cell reaction in CeD. Here, major immunogenic gluten peptides are incompletely degraded by mammalian proteases and thus reach the intestinal lamina propria, the site where activation of gluten-specific T cells occurs. This review mainly focuses on the current status of an expanding spectrum of gluten-degrading enzymes and their perspectives as potential therapeutic agents for an improved management of CeD.

## 2. Origin and Properties of Gluten

Gluten proteins are found in wheat, which belongs to the *Triticeae* tribe in the *Poaceae* plant family [8]. Wheat is one of the most widely consumed cereals in the world and is the greatest single source of protein in the human diet. Together with maize and rice, wheat species account for over 70% of the total cereal production worldwide [9,10]. Due to their abundance, widespread distribution and consumption amongst human and animal populations, these cereals are of great nutritional and economic importance.

The structure and properties of cereal grains have been studied systematically for over 250 years. The protein content of gluten from various wheat species was among the first to be characterized. In 1745, Beccari first described the isolation of wheat gluten [11]. Two centuries later, Osborne developed a classification system for plant-derived proteins based on their extraction and solubility in different solvents [12]. This classification comprised the albumins (water soluble), globulins (saline soluble), glutelins (dilute acid or alkali soluble) and prolamins (alcohol-water soluble).

Prolamins are highly heterogeneous and represent the major seed storage protein of cereal grains such as wheat (gliadin), barley (hordein), rye (secalin), corn (zein), sorghum (kafirin) and oats (avenin). Their molecular mass can range from 10 to 100 kDa. Prolamins of the *Triticeae* family (wheat, barley, rye) are classified into one of three groups: sulphur-rich (S-rich), sulphur-poor (S-poor) and high molecular weight (HMW) prolamins, with several sub-groups contained within the S-rich group [13,14]. One distinct structural feature of prolamins is the presence of amino acid sequences consisting of repeated short peptide motifs. Another remarkable characteristic of these proteins is the high content of specific amino acids, especially proline and glutamine. Hence their name prolamin is based on the fact that these proteins are rich in proline and the (amide) nitrogen derived from glutamine.

## 3. Gluten Structure

Gluten proteins comprise roughly 80–90 percent of the total wheat endosperm (kernel) protein. About 35% of their amino acid content is made up of glutamine and 15% of proline residues [8,15]. Due to this unique amino acid composition, combined with a high percentage of hydrophobic amino acids (19%), certain protein sequences in gluten are highly resistant to degradation by human gastrointestinal proteases [16,17], including pepsin, trypsin, chymotrypsin, carboxypeptidases A and B, elastases and enzymes of the small intestinal brush-border membrane. The reason for the inefficiency of these proteases to degrade these gluten sequences is the general lack of cleavage-site activity for intra-chain post-proline residues.

Gluten confers the desired dough making and baking qualities to wheat, as well as much of the gustatory uniqueness of wheat-based foods. This is largely due to gluten resembling a polymer. In gluten, disulfide linked polymers (via intermolecular S-S-bonds) of glutenins intercalate with single gliadin molecules [8,15], forming a protein network. The network traps carbon dioxide and water during dough fermentation which are retained after baking.

Gliadins are proteins without a delicate secondary structure accounting for 40–50% of the total storage proteins of wheat. They can be separated into four discrete electrophoretic fractions and classified as α, β, γ and ω-gliadins [13]. Based on their primary structure and molecular weight, gliadins have also been classified as ω1, 2-, ω5-, α/β- and γ-gliadins. The α/β-gliadins have compact globular structures, and γ- and ω-gliadins have extended rod-like structures [18]. The *α*/*β*- and *γ*-gliadins harbor three and four intramolecular (SS) bonds, respectively, while *ω*-gliadins lack cysteine residues. All gliadins harbor T cell stimulatory peptides, but two peptides derived from α- and γ-gliadin, a 33mer and a 26mer, respectively, are particularly strong activators of T cells and therefore highly correlated to the onset and development of CeD [16,19,20] (Figure 1).

## 4. Gluten Immunogenic Peptides as Drivers of CeD

Gliadins and glutenins harbor at least 50 different immunogenic peptide sequences that are largely resistant to gastrointestinal peptidases [3,15,21,22]. These peptides can reach the small intestinal lamina propria, in part through active transepithelial transport. In the lamina propria they are antigenically presented on macrophages, dendritic and B cells in the context of the human leukocyte antigens (HLA)-DQ2 and HLA-DQ8. Notably, HLA-DQ2 or HLA–DQ8 are a necessary but not a sufficient precondition for CeD, since about one third of most human populations carry at least one of these HLA-genes. The CeD autoantigen tissue transglutaminase (TG2), which is released from most cells in conditions of stress, [23,24,25,26] can deamidate the glutamine residues or crosslink them to a lysine residue of another protein. Deamidation enhances the peptides’ fit into the antigen binding groove of HLA-DQ2 or HLA-DQ8, and crosslinking promotes their uptake and processing by the antigen presenting cells [22,25]. Both these TG2-mediated alterations of the gluten peptides that enter the lamina propria are therefore a central step towards activation of gluten-specific T cells, with subsequent intestinal inflammation, villous atrophy and reactive crypt hyperplasia.

The immunogenicity of these peptides in patients with CeD depends on their primary structure, which permits a low to medium affinity fit into the antigen binding groove of HLA-DQ2 or HLA–DQ8 that is governed by the direct interaction of the nine amino acid linear core sequence of the gluten peptide and its interaction with the amino acids of the HLA molecule in which it becomes embedded. As illustrated in Figure 2, TG2-mediated deamidation of certain glutamines in the immunogenic gluten peptides produces a negative charge by creating a glutamic acid residue that increases the otherwise modest binding affinity.

The other essential precondition for effective antigenic presentation of these gluten peptides in the intestinal lamina propria is their partial resistance to degradation by mammalian gastrointestinal proteases. This resistance is due to their high content in proline residues, a unique feature of wheat prolamins [8]. Consequently, during passage through the upper gastrointestinal tract, gluten proteins are cleaved into smaller peptides, but several larger (10–40 amino acids length) fragments remain undigested. Here, the two larger gliadin-derived 26- and 33-mer peptides (the γ-gliadin 26-mer sequence FLQPQQPFPQQPQQPYPQQPQQPFPQ [28], and the α2-gliadin 33-mer sequence LQLQPFPQPQLPYPQPQLPYPQPQLPYPQPQPF [16]) are especially noteworthy for their strong induction of destructive T cell responses in patients with CeD (Table 1 and Figure 1).

Both peptides contain several (overlapping) immunogenic T cell epitopes (Table 1) that can bind top HLA-DQ2 and HLA-DQ8, especially after TG2-mediated deamination [15]. These and several other peptides (Table 1) are subsequently recognized and activated by the respective gluten peptide-specific CD4^+^ T-cells of the intestinal lamina propria, to elicit a type 1 helper cell (Th1) response. Th1 cells secrete interferon gamma (IFN-γ) and other pro-inflammatory cytokines.

They also contribute to the B cell-mediated humoral response against the select gluten and deamidated gluten peptides, as well as to (otherwise cryptic) epitopes of the autoantigen TG2 itself [3,6,30]. Thus, CeD can be considered an auto-destructive disease that is maintained with the continuous supply of gluten (Figure 3).

## 5. The Approach for the Use of Gluten-Degrading Enzymes in CeD

There is a great interest among CeD patients in a medical therapy that can ameliorate gluten-induced effects. Such therapy is likely not able to “neutralize” the large amounts of gluten (10–30 g per day) in an average wheat-based diet. Rather, a pharmacological therapy should eliminate the effect of up to a few grams of gluten in a largely gluten-free diet (GFD) that avoids overt gluten sources, such as bread, pizza, pasta or cookies. Most CeD patients agree that such a diet would be much more sustainable, while the needed strictly gluten free diet (less than 20 ppm of gluten in all foods, i.e., less than 20 mg/kg) is a great challenge in everyday life [31]. Moreover, in a double blind clinical study, CeD patients in remission who were challenged with 50 mg gluten daily developed a 20% decrease in villous height/crypt depth vs placebo or 10 mg gluten daily [32], indicating that such minor amounts of gluten may cause chronic mucosal damage. In addition, high sensitivity to minor amounts of gluten may underly refractory celiac disease type 1. A major therapeutic approach to such supportive therapy is the use of bacterial of grain-derived gluten-degrading enzymes, first spearheaded by the group of Khosla [28,30,33,34,35,36] while commercially available “glutenases” have no or little proven efficacy to effectively degrade antigenic gluten peptides [37].

The search for alternate therapeutic pathways for the treatment of CeD has become a topic of high importance for the celiac community and health care providers [30,38,39]. Food-grade proteases capable of detoxifying moderate quantities of dietary gluten (up to 3 g gluten per day in view of an average daily consumption of 15–20 g gluten) [40] could help mitigate this problem [34].

The employment of glutenases, i.e., proline and glutamine-specific endoproteases, as therapeutic agents has been proposed as a viable new therapy for the treatment of CD [30,36,41,42]. Since mammalian gastrointestinal proteases lack the capability to efficiently cleave and abolish the immunogenic properties of the relevant gluten peptides, exogenous sources of proteolytic enzymes are being explored for their potential to complement the human-encoded enzymatic repertoire, to digest the otherwise proteolysis-resistant peptides [28,42,43,44]. Logically, this may be achieved by the ingestion of exogenous endoproteases, which must invariably be able to cleave and neutralize the immunogenic gliadin and glutenin epitopes quickly enough before an inflammatory response is elicited in the proximal small intestine of patient. If this purpose is fulfilled, then these endoproteases may be considered “true” therapeutic candidates for the treatment of CeD.

## 6. Classification and Origin of Gluten-Degrading Enzymes

A variety of different gluten-degrading enzymes have been identified in bacteria, fungi and plants. For instance, barley produces a seed-derived glutamine-specific cysteine endoprotease, and some bacteria produce prolyl endopeptidases or subtilisins, which are both very effective in digesting gluten. Some of the most promising enzymes that are pre-clinical or have been considered for clinical trials are discussed below.

### 6.1. Prolyl Endopeptidases (PEP)

Prolyl endopeptidases (PEP) are the enzymes that target the typical proline-rich regions of gluten, several of which harbor T cell immunogenic epitopes. PEP are derived from bacteria and fungi.

Bacterial PEP are derived from *Flavobacterium meningosepticum* (FM-PEP), *Sphingomonas capsulata* (SC-PEP, or ALV002) and *Myxococcus xanthus* (MX-PEP) [28]. The physicochemical and biochemical properties of FM-PEP and SC-PEP have been well characterized before consideration as therapeutics for CeD [45,46,47].

In a comparative study, the gluten degrading properties of FM-PEP, SC-PEP and MX-PEP were evaluated using two substrates, the 33-mer super-epitope (LQLQPFPQPQLPYPQPQLPYPQPQLPYPQPQPF) and the related but smaller sequence PQPQLPYPQPQLP (a recurring immunogenic sequence derived from several α-gliadins). PEP preferentially cleave P-Q bonds that are usually found in immunogenic gluten peptides and that are resistant to mammalian gastrointestinal proteases (Figure 4). These peptides have been widely recognized as useful probes for the study of both the proteolytic properties of PEP and for the assessment of proteolytic gluten inactivation in CeD [16,19]. The cited studies showed that although all three enzymes tested showed highly specific activity against reference chromogenic substrates, they exhibited differences in specificity depending on the gluten peptide chain-length. SC-PEP and MX-PEP showed higher specificity towards PQPQLPYPQPQLP, whereas FM-PEP preferentially cleaved the longer 33-mer. In contrast, SC-PEP showed poor activity towards the 33-mer. All three enzymes were structurally stable and active at pH 6-7, a pH activity profile that is well suited for the mildly acidic environment of the upper small intestine, the anatomical region where CeD occurs. Furthermore, these enzymes showed moderate stability toward pancreatic proteases and acid exposure, but their proteolytic capabilities were completely inactivated by pepsin, requiring their protection from gastric acid and pepsin. In vivo, these PEP retained their proteolytic activities in the small intestinal lumen of rats, which suggested that these enzymes were resistant to inactivation by brush border membrane peptidases.

A PEP is also produced by the fungus *Aspergillus niger* (AN-PEP or ASP) [44,48]. This enzyme was first identified and described as glutenase by Edens and coworkers in 2005 [49], who discovered it while pursuing a debittering agent for casein hydrolysates and for applications in beer brewing. Among the advantageous properties of AN-PEP is that the enzyme exhibits a high activity towards intact gluten and that it eliminated the T-cell stimulatory properties of a peptic/tryptic digest of gluten that mimics aspects of human digestion, but otherwise fails to inactivate major immunogenic gluten epitopes [44,48]. Furthermore, as opposed to the bacterial PEP derived from SC, FM and MX, which presented an optimum pH activity profile between 7.0 and 8.0 (and therefore cannot function in the acidic environment of the stomach), AN-PEP is active at lower pH values. The optimum activity was determined between pH values of 4.0 and 5.0. This finding led to the conclusion that AN-PEP may be active in the stomach, hence potentially removing the necessity of employing an enteric coating, and allowing early-on digestion of gluten already in the stomach, increasing the time window in which the enzyme can deactivate with the ingested gluten peptides. However, three small phase 1 clinical trials showed insufficient gluten degrading activity by AN-PEP in healthy controls and gluten challenged patients with celiac disease in remission [50,51,52].

### 6.2. Glutamine-Specific Cysteine Endoprotease (EP-B2)

Other studies have addressed the proteolytic capabilities of a glutamine-specific cysteine endoprotease derived from seeds of germinating barley (EP-B2, or ALV001) [41,53,54]. An in vivo study revealed that clinically achievable oral doses of the EP-B2 proenzyme, but not placebo, could ameliorate clinical relapse in a gluten-sensitive macaque [55]. Khosla’s team performed studies on isolation, upscaling and in vivo validation of EP-B2. Thus, they isolated a proenzyme of EP-B2 by heterologous expression in *Escherichia coli* [53] and generated the zymogen forms of EP-B2 from gram to kilogram quantities through a scalable fermentation, refolding and purification process [56]. As mentioned before, gluten-derived peptides contain a high proportion of proline and glutamine residues within their structure. Since EP-B2 has cleavage site specificity for post-glutamine residues, a combination enzyme therapy of both EP-B2 and PEP, which can effectively cleave these peptides after proline residues, has been developed for the treatment of CeD. Thus, in vitro studies have shown that combinations of EP-B2 with EM-PEP or SC-PEP were able to synergistically digest gluten peptides and effectively reduce their immunogenicity under simulated gastric conditions [35,41]. The gluten-degrading efficacy of EP-B2 is also enhanced by combination with AN-PEP under simulated gastric conditions [34]. Therefore, EP-B2 is still considered a therapeutically promising oral protease when used in combination with other peptidases.

### 6.3. Gluten-Degrading Enzymes in Human Saliva

Human saliva is a fluid comprising various contributors both from exocrine and non-exocrine origin. While glandular salivary fluid secretions are sterile in nature, the oral cavity itself is richly colonized with a wide variety of different oral microbial species. The bacteria primarily belong to the Actinobacteria and Bacteroidetes phyla [57]. Since gluten when ingested first come into contact with the bacteria present in oral fluid, saliva and the bacteria it contains are of great interest from the perspective of potential gluten-degrading activities [58]. It was found that saliva contains glutamine endoprotease activity and that the enzymes are of microbial origin [59]. Subsequently, it was discovered that the microbes from saliva and dental plaque are also capable of degrading gluten, including the highly immunogenic 33-mer and 26-mer gliadin sequences (Table 1) [42]. These findings raised interest in the oral microbe-derived gluten-degrading enzymes [60,61,62,63]. The specific gluten-degrading bacteria were isolated from among the over 1200 different oral microbial species present in saliva with a gluten-limited agar approach [64], which led to the identification of *Rothia* species (*Rothia aeria and Rothia mucilaginosa*) as the most effective gluten-degrading oral microorganisms [42,43]. The enzyme from *Rothia mucilaginosa* was subsequently successfully isolated and identified as belonging to the S8 family of peptidases called subtilisins. The food-grade Bacillus species, which also produce subtilisins, were likewise shown to cleave and abolish gluten immunogenic epitopes. *Rothia* species cleave both the R5 epitope from ω- gliadin (sequence QQPFP) and the G12 epitope from α-gliadin (sequence LGQQQPFPPQQPY) that are contained in core immunogenic gliadin sequences and that represent central epitopes for determination of gliadin immunogenicity in flours and food products [59,60,61,62,63,65] This suggests that the subtilisins produced by *Rothia bacteria* theoretically eliminate most if not all immunogenicity associated with gliadins. Subtilisins thus represent an as yet understudied class of enzymes with great potential for enzyme therapeutic applications in CeD [62], especially since they already have food-grade status.

## 7. Challenge of Glutenase Therapy for Celiac Disease

Gluten-degrading enzymes, can both increase and decrease gluten peptide immunogenicity, as illustrated by peptidases and prolyl endopeptidases produced by commensals in the human GI tract. The increase of immunogenicity can be explained by the generation of gluten peptide fragments that more easily enter the small intestinal lamina propria than the larger immunogenic peptides produced by the human gastrointestinal proteases. An example is an enzyme produced by *Pseudomonas* that increased the number of smaller immunogenic peptides, while only the additional activity of the Lactobacillus protease then leads to smaller non-immunogenic gluten fragments [37]. Therefore, therapeutic glutenases must be able to rigorously degrade immunogenic epitopes before they reach the proximal small intestinal mucosa.

Further, structural instability of gluten-degrading enzymes in the digestive tract, and autodegradation of enzymes, are major challenges for the therapeutic application of these enzymes [65,66]. Due to the proteinaceous nature of enzymes, they are subject to inactivation and/or digestion in the gastrointestinal tract by gastric acid and under proteolytic condition. As a consequence, these enzymes may fail to degrade and efficiently inactivate gluten before it reaches the small intestine, the site where gluten induces inflammatory T cell responses that lead to CeD. As mentioned above, AN-PEP degrades gluten proteins under acid conditions in vitro and therefore appeared to qualify as an oral supplement to reduce gluten exposure in patients [44]. However, glutenases that are currently marketed as dietary supplements, including AN-PEP, have not been demonstrated to sufficiently degrade gluten in vivo and are therefore not to be recommended as a supplement [37].

## 8. Novel Strategies for Enzyme Therapies

When enzymes are to be utilized to aid in the digestion of gluten, some additional challenges need to be faced. Enzymes need to degrade gluten to the point where immunogenicity is abolished before the gluten-containing food, which is usually embedded in a complex food matrix, reaches the proximal small intestine. Further, enzymes are proteins and are likely susceptible to proteolytic degradation in the harsh environment of the gastrointestinal tract, which is highly enriched in proteolytic enzymes, not the least being pepsin, trypsin and chymotrypsin. Lastly, gluten-degrading enzymes, such as subtilisins, have a defined secondary structure, which is susceptible to denaturation and hence abolishment of enzymatic activity at acidic pH, as encountered in the stomach.

Several strategies have been pursued to overcome the above critical issues: First, to use combinations of enzymes having different optimum pH values to optimize digestion and to achieve activity under low as well as neutral pH conditions; second, to use molecular modeling to generate a new enzyme by starting from an acidic protease and by attempting to further increase its specificity and especially activity for gluten; third, to chemically modify the enzyme by encapsulation and/or attaching groups such as poly-ethylene-glycol (PEG), to reduce protease susceptibility and/or pH sensitivity. The results of these targeted enzyme optimizations are described below.

## 9. Enzyme Combinations

Potential synergism between gluten-degrading enzymes that differ in their cleavage specificities raises the possibility that a combination would more effectively eliminate the antigenicity of ingested gluten fractions in vivo before these peptides could reach the duodenal lumen. The combination of EP-B2 (ALV001) and SC-PEP (ALV002) in a 1:1 ratio (ALV003, also known as Latiglutenase [35]) showed promising results in phase 1b-2a clinical studies, demonstrating significant protection from the development of villous atrophy and intraepithelial lymphocytosis in CeD subjects in remission who were challenged with the enzymes plus gluten versus placebo plus gluten [67,68]. However, a “real-life”, 500-patient multicenter trial where patients on a strict GFD who still exhibited villous atrophy and abdominal complaints related to CeD were randomized to receive the combined enzymes at different doses versus placebo for 12 or 24 weeks failed to demonstrate a benefit, since both the placebo and the enzyme-treated groups showed a slight improvement [69]. Apart from the overall lack of efficacy of ALV003 on intestinal histology, this study indicates that in patients on a strict GFD, the clinical study setting can further improve diet compliance and therefore small intestinal histology. Despite these disappointing results, the enzyme combination still has potential as an adjunctive therapy to other pharmacological approaches. Thus, a recent sub-analysis of the larger trial showed that the subgroup of patients with TG2-autoantibody positivity had an improvement of symptoms and quality of life with the enzyme combination [70].

Another combination that has been explored is the combination of aspergillo-pepsin from *Aspergillus niger* (ASP) and dipeptidyl peptidase IV (DPP-IV), from *Aspergillus oryzae*. The combination was found to successfully degrade small amounts of gluten in vitro [34]. ASP and DPP-IV are widely used in the food and feed industry as dietary supplements. Neither ASP nor DPPIV efficiently cleaved synthetic immunogenic gluten peptides when used alone, but when ASP was combined with DPPIV or with EP-B2 (ALV-002, as component of ALV-003), it exhibited only modest to moderate activity to reduce immunogenic gluten [34]. However, to date, ALV-003 remains the most effective clinically tested enzyme combination that still has potential as adjunctive therapy in CeD.

## 10. Molecular Modeling

A promising approach has been to design an enzyme that is active under acidic conditions and to introduce, through genetic modification, the desired substrate specificity. Using molecular modeling, an acid-active endopeptidase from the acidophilic bacterium *Alicyclobacillus sendaiensis* was genetically altered to facilitate the digestion of gluten [71]. The original enzyme from *Alicyclobacillus sendaiensis*, called kumamolisin-As (KumaWT), is a serine endoprotease, which exhibits an optimal activity at pH 2.0-4.0 [71]. The engineered KumaWT enzyme, called KumaMax or Kuma010, exhibited a 116-fold higher proteolytic activity, and an 877-fold higher specificity for a set of target gliadin oligopeptides than KumaWT. The enzyme was then further altered to increase activity towards the immunogenic 33-mer and 26-mer peptides [71]. This enzyme, called Kuma030, was more efficient in vitro than ALV003; at a 1:10 enzyme to substrate (w/w) ratio, gluten was degraded by 84% by ALV003, while Kuma030 at an enzyme to substrate ratio of 1:40 degraded over >99.9% of these key immunogenic gliadin peptides lowering the gluten content to about 3 ppm, which is below the 20 ppm cut-off value for gluten-free food labeling. When gliadin-specific T cells were exposed to Kuma030-digested T cell clones, T IFN-γ production and proliferation was reduced to almost zero in a dose-dependent manner [72]. Currently, Kuma030 is tested in a phase 1-2 clinical trial as a novel enzyme therapeutic for CeD (NCT03409796).

## 11. Pharmaceutical Enzyme Modification by PEGylation and Microencapsulation

Protein-based drugs hold promise as therapeutic agents because of their high specificity, but in general they often display short half-lives in vivo, either due to rapid excretion or proteolysis. Instability and auto-degradation of gluten-degrading enzymes are major challenges for their therapeutic application.

PEGylation usually improves protein stability [73]. PEGylation was first described in 1977 as a valuable technique to stabilize catalase and albumin [74]. PEGylation consists of covalent or non-covalent attachment of poly-ethylene-glycol (PEG) to a molecule such as a therapeutic protein or to a nanocarrier, thereby enhancing its protection from proteolytic degradation or rapid excretion, thus increasing its stability in vivo [73,75]. Microencapsulation is a technique of enteric coating to form microparticles that partly resist enteric degradation or allow distal delivery of the pharmacological agents or protein. Examples of biocompatible polyesters are poly (lactic-co-glycolic acid) copolymers (PLGA). As a polymeric vehicle, predictability of degradation, ease of fabrication, and regulatory FDA approval are features that make PLGA desirable for medical applications [76].

The gluten-degrading enzyme subtilisin-A (Sub-A), like many other enzymes, is only weakly active and unstable under acidic conditions [77]. PEGylation in combination with PLGA (polymeric lactic-glycolic acid) microencapsulation was found to effectively protect the gluten degrading enzyme subtilisin-A from inactivation due to acid exposure, as well as from autoproteolysis [65]. The gluten degrading subtilisin-A from *B. licheniformis* (Sub-A) was modified by PEGylation and microencapsulation as schematically represented in Figure 5 [65].

When unprotected, Sub-A activity was lost to a large extent at low pH (1.5–3.0) and could not be recovered after adjustment of the pH to neutral pH [65]. Methoxypolyethylene glycol (mPEG, 5 kDa) was used to modify Sub-A, to form PEGylated Sub-A (Sub-A-mPEG). It was demonstrated that this modification protected Sub-A from autolysis at neutral pH. The PEGylated Sub-A (Sub-A-mPEG) was further encapsulated by PLGA. The microencapsulated Sub-A-mPEG-PLGA showed significantly increased protection against acid exposure. Protection was demonstrated in vitro, and the effectiveness was confirmed in vivo [65]. These results show that pharmaceutical modification can protect Sub-A, and likely other glutenases, from auto-digestion as well as from acid inactivation in the stomach, thus rendering the enzyme more effective for applications in vivo.

## 12. Summary

Immunogenic gluten peptides that survive gastrointestinal degradation are the decisive triggers for CeD. Using glutenases to abolish their immunogenic potential is an attractive option for patients with CeD. Bacterial, fungal and plant derived glutenases, especially (food-grade) subtilisins (Sub-A), prolyl endopeptidases (PEP), barley seed derived glutamine-specific cysteine endoprotease (EP-B2) and synthetic glutenases (Kuma030) are considered promising candidates for an (adjunctive) oral enzyme therapy. Enzyme instability and autodegradation of enzymes are challenges for their therapeutic application but are actively being addressed. Successful strategies that are currently being explored are enzyme combinations, molecular modeling and chemical modifications such as PEGylation, enteric coating of enzymes or enzyme carriers. Latiglutenase, KumaMax, and modified subtilisins modified by PEG and in combination with PLGA microencapsulation are more recent developments that may yield formulations that can be used as to reduce and abolish immunogenicity if ingested gluten for the management of CeD. A major challenge for enzyme therapy remains to secure rapid and complete enzymatic digestion of immunogenic gluten peptides that are embedded in a complex food matrix. This must occur within the stomach and proximal small intestine before these peptides reach the mucosal immune system of the small intestine.

## Figures and Tables

**Figure 1 nutrients-12-02095-f001:**
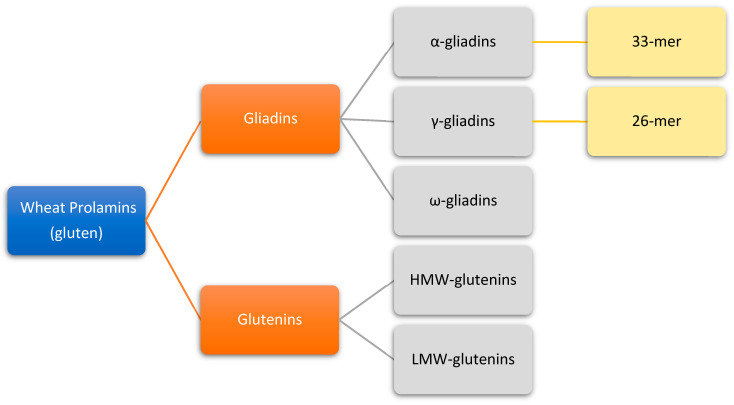
Classification of wheat prolamins (gluten). The α- and γ-gliadins harbor peptides with prominent immunogenicity for CeD patients (33-mer and 26-mer).

**Figure 2 nutrients-12-02095-f002:**
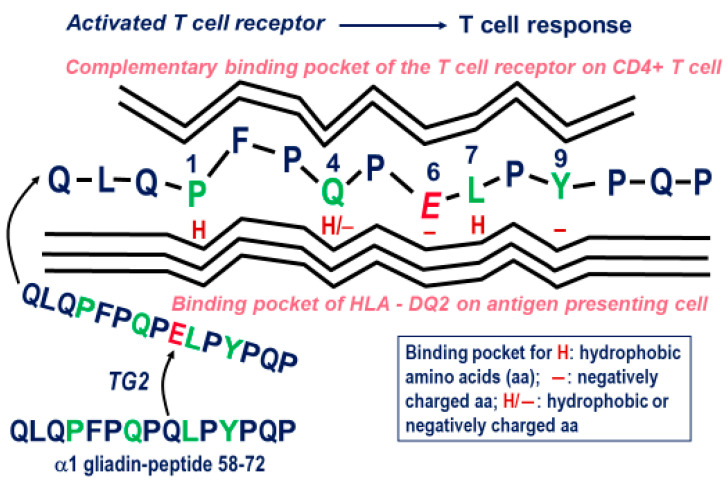
Activation of a gluten-specific T cell by an immunogenic gluten peptide. The affinity of the peptide to HLA-DQ2 is highly increased by TG2-mediated deamidation of glutamine residue (Q) in position 9 of the 15-mer peptide, to yield an acidic glutamic acid residue (E) that shows improved binding to the antigen binding groove of HLA-DQ2 on the antigen presenting cell (macrophage, dendritic or B cell). HLA-DQ2 requires a core consensus sequence of nine amino acids, with preference of hydrophobic or negatively charged amino acids in the gluten peptide. Modified from Schuppan et al. [3,27].

**Figure 3 nutrients-12-02095-f003:**
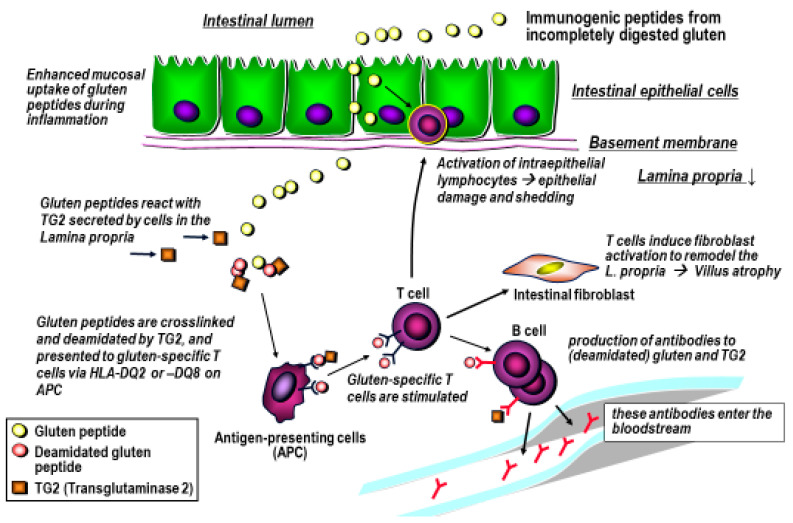
Pathomechanism of celiac disease. Simplified scheme depicting the intestinal uptake of incompletely digested immunogenic gluten peptides that, after deamidation by TG2, bind to antigen presenting cells in the lamina propria, to elicit a strong T cell mediated immune response. The consequence is remodeling of the lamina propria and destruction of the resorptive villi. The activation of B cells that produce antibodies to (deamidated) gliadin and the autoantigen TG2 is maintained as long as T cells are stimulated by the nutritional supply of gluten. Modified from Schuppan et al. [3,27].

**Figure 4 nutrients-12-02095-f004:**
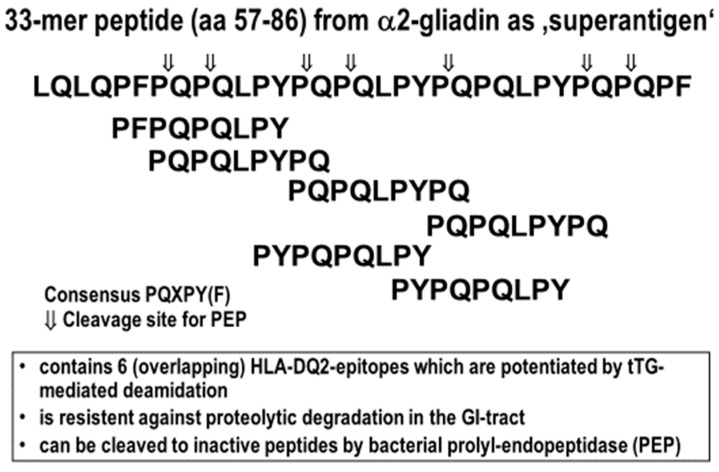
Cleavage sites of PEP in the otherwise gastrointestinal protease resistant highly immunogenic α2-gliadin 33-mer. The six similar, overlapping immunogenic HLA-DQ2 binding peptide epitopes within the 33-mer are illustrated. Cleavage sites are derived from [16,19].

**Figure 5 nutrients-12-02095-f005:**
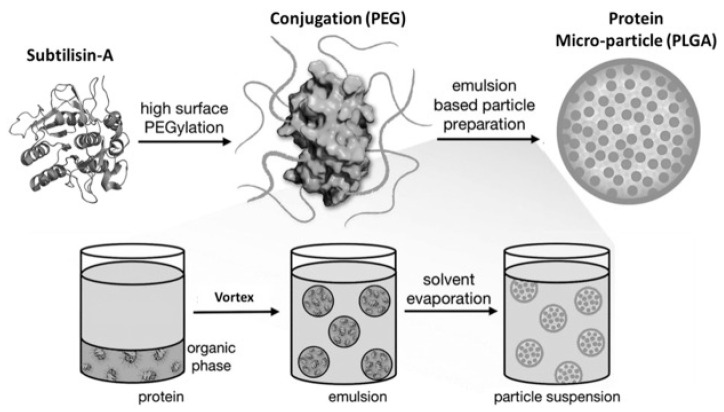
Schematic representation of the pharmaceutical coating procedure applied to Sub-A. Reproduced with permission [71].

**Table 1 nutrients-12-02095-t001:** Selection of potent T-Cell stimulatory epitopes.

Peptide	Amino Acid Sequence	HLA	tTG
**Gliadins:**			
Glia α (206–217)	SGQGSFQPSQQN	DQ8	(+)
Glia-α2 (62–75)	PQPQLPYPQPQLPY	DQ2	(+++)
Glia-α2 33mer (56–88)	LQLQPFPQPQLPYPQPQLPYPQPQLPYPQPQPF	DQ2	(+++)
Glia-α9 (57–68)	QLQPFPQPQLPY	DQ2	(+++)
Glia-α20 (93–106)	PFRPQQPYPQPQPQ	DQ2	(+++)
Glia- γ 1 (138–153)	QPQQPQQSFPQQQRPF	DQ2	(+++)
Glia- γ(5) 26mer (26–51)	FLQPQQPFPQQPQQPYPQQPQQPFPQ	DQ2	(+++)
Glia- γ 30 (222–236)	VQGQGIIQPQQPAQL	DQ2	(-)
**Glutenins:**		DQ2	(+++)
LMW-Glt-156 (40–59)	QPPFSQQQQSPFSQ	DQ2	(+++)
LMW-Glt-17 (46–60)	QQPFSQQQQQPLPQ	DQ2	(+++)
LMW-Glt (723–735)	QQGYYPTSPQQSG	DQ2	(+++)
		DQ2	(+++)
Glu-5	QQQXPQQPQQF	DQ2	(+++)
Glu-21	PQQSEQSQQPFQPQ	DQ2	(---)

Sequences have been taken from gliadins and glutenins and residues that can be deamidated by TG2 are highlighted in Red. Modified from [15,19,20,29].

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
