# Peer review of "Gluten Degrading Enzymes for Treatment of Celiac Disease"

_nutrients, 2020, doi:10.3390/nu12072095_

Round 1

Reviewer 1 Report

I think that this is an excellently written review and a valuable and timely contribution to the literature.

Author Response

We thank the reviewer for his/her positive assessment.

Reviewer 2 Report

Very interesting paper; Gluten properties and immunogenicity sections are very well described.

A few comments:

  • Ref #4 for institution is missing.
  • Several assertions with no references: line 62; line 73; line 74.
  • Line 184-185: the sentence “a sizable percentage of CeD patients is highly sensitive to even minor amounts of gluten” is disputable. What percentage? What means “highly sensitive individuals” and “minor amounts”? Please clarify and provide references.
  • Line 191-92: About the sentence “Food-grade proteases capable of detoxifying moderate quantities of dietary gluten could help mitigate this problem”, please indicate what authors mean by “moderate quantities”
  • lines 217 and 224: for the study to which the text refers, the reference is missing.
  • A few typos: lines 220, 242, 334
  • Lines 305-307: I presume it is meant Enzymes produced by Pseudomonas
  • A sentence about current challenges and main obstacles to future therapeutic application should also be mentioned in the summary (last section)

Author Response

Ref #4 for institution is missing.

Response: thank you; there are indeed only 3 institutions – this has been corrected.

Several assertions with no references: line 62; line 73; line 74.

Response: Thank you, References have been added now (lines 63, 73, and 75).

Line 184-185: the sentence “a sizable percentage of CeD patients is highly sensitive to even minor amounts of gluten” is disputable. What percentage? What means “highly sensitive individuals” and “minor amounts”? Please clarify and provide references.

Response: We changed and specified our statement as follows (including the resp. citation): “Moreover, in a double blind clinical study CeD patients in remission who were challenged with 50 mg gluten daily developed a 20% decrease in villous height/crypt depth vs placebo or 10 mg gluten daily (novel ref#32), indicating that such minor amounts of gluten may cause chronic mucosal damage. In addition, a high sensitivity to minor amounts of gluten may underly refractory celiac disease type 1. “

Line 191-92: About the sentence “Food-grade proteases capable of detoxifying moderate quantities of dietary gluten could help mitigate this problem”, please indicate what authors mean by “moderate quantities”

Response: In most current clinical studies testing pharmacological agents for CeD, an upper limit of 3 g gluten per day is considered the standard dose in view of an average daily consumption of 15-20 g gluten daily. We therefore specified this as follows and inserted a reference: “Food-grade proteases capable of detoxifying moderate quantities of dietary gluten (up to 3 g gluten per day in view of an average daily consumption of 15-20 g) (ref#40) could help mitigate this problem.”

lines 217 and 224: for the study to which the text refers, the reference is missing.

Response: The study refers to cleavage sites by PEP which preferentially cleaves P-Q bonds. References 13 and 16 have been added.

A few typos: lines 220, 242, 334

Response: Thank you, these have been corrected.

Lines 305-307: I presume it is meant Enzymes produced by Pseudomonas …

Response: Yes, this has been corrected.

A sentence about current challenges and main obstacles to future therapeutic application should also be mentioned in the summary (last section)

Response: We have added the following sentence at the end of the summary: “A major challenge for enzyme therapy remains: to secure rapid and complete enzymatic digestion of immunogenic gluten peptides that are embedded in a complex food matrix. This must occur within the stomach and proximal small intestine before these peptides reach the mucosal immune system of the small intestine. “

Reviewer 3 Report

As an overview of gluten degrading enzymes this review is potentially of interest to readers. The Abstract & Introduction are not well written in places. Abstract: 1st line (16-18) reads poorly. Second line (18-20) all people ingesting gluten will have proteins resistant to degradation-the sentence is meaningless. The next few lines are 'clunky'. Overall the abstract could be more focused. The abstract should comment on the clinical experience and direction of travel for enzymes as a therapy.

Introduction: first line (44-45) again messy. Throughout very little mention of inflammation e.g. line 3 (46-47), which is the driving force for the histological changes & clinical disease. Line (47-50) is a mess. In addition the authors are not consistent, lots if not most patients with celiac are asymptomatic of have mild disease (56-59). Most patients would not consider diarrhoea or malabsorption mild symptoms. Only a small percentage are severely symptomatic or are highly sensitive to small traces of gluten (this is also mentioned line 184-186). Line 60-62, why use travels as an example, seems a bit random, there are a multitude of pressures influencing peoples behaviour. Only a small percentage of patients are symptomatic when compliant.

Summary: Need to mention clinical experience to date.

Author Response

As an overview of gluten degrading enzymes this review is potentially of interest to readers. The Abstract & Introduction are not well written in places. Abstract: 1st line (16-18) reads poorly. Second line (18-20) all people ingesting gluten will have proteins resistant to degradation-the sentence is meaningless. The next few lines are 'clunky'. Overall the abstract could be more focused. The abstract should comment on the clinical experience and direction of travel for enzymes as a therapy.

Response: We have changed the first sentence of the abstract as follows “Celiac disease (CeD) affects about 1% of most world populations. It presents with a wide spectrum of clinical manifestations, ranging from minor symptoms to mild or severe malabsorption, and it may be associated with a wide variety of autoimmune diseases.”

The following sentence was modified as follows: “CeD is triggered and maintained by the ingestion of gluten proteins from wheat and related grains. Gluten peptides that resist gastrointestinal digestion are antigenically presented to gluten specific T cells in the intestinal mucosa via HLA-DQ2 or HLA-DQ8, the necessary genetic predisposition for CeD.”

To refer more distinctly to clinical evaluation and clinical studies, we have changed the last sentence of the abstract as follows: “A This review focuses on those enzymes that have been characterized and evaluated for the treatment of CeD, discussing their origin and activities, their clinical evaluation and challenges for therapeutic application. Novel developments include strategies like enteric coating and genetic modification to increase enzyme stability in the digestive tract.”

Introduction: first line (44-45) again messy. Throughout very little mention of inflammation e.g. line 3 (46-47), which is the driving force for the histological changes & clinical disease. Line (47-50) is a mess. In addition the authors are not consistent, lots if not most patients with celiac are asymptomatic of have mild disease (56-59). Most patients would not consider diarrhoea or malabsorption mild symptoms. Only a small percentage are severely symptomatic or are highly sensitive to small traces of gluten (this is also mentioned line 184-186). Line 60-62, why use travels as an example, seems a bit random, there are a multitude of pressures influencing peoples behaviour. Only a small percentage of patients are symptomatic when compliant.

Summary: Need to mention clinical experience to date.

Response: We do not fully understand all points of this critique: Thus, CeD is characterized as an inflammatory disease in the first line of Introduction (line 44). However, we have edited central passages, to address all points raised, with now clearer statements.

We have now specified the inflammatory T cell response in line 53 as follows: “…..,which response drives an inflammatory T helper 1 (Th1) cell response resulting in villous atrophy and usually in clinical disease.”

We have changed lines 58-60 as follows: “One reason for this is that nowadays most patients with active CeD have no or only minor abdominal symptoms such as diarrhea or overt malabsorption, despite concomitant extraintestinal autoimmunity and an increased risk for malignancy.”

Lines 185-89 were changed also according to comments by reviewer #2: “Moreover, in a double blind study CeD patients in remission who were challenged wit 50 mg gluten daily developed a 20% decrease in villous height/crypt depth vs placebo or 10 mg daily (Ref#32) , indicating that such minor amounts of gluten may cause chronic mucosal damage. In addition, high sensitivity to minor amounts of gluten may underly refractory celiac disease type 1.

Re the mentioning of travels: we used the phrase “…..in our societies or during travels”, which should include other societal pressures.

Re the summary, also in response to reviewer #2, we have added the following sentence: “A major challenge for enzyme therapy remains: to secure rapid and complete enzymatic digestion of immunogenic gluten peptides that are embedded in a complex food matrix. This must occur within the stomach and proximal small intestine before these peptides reach the mucosal immune system of the small intestine. “
